# Characterization of a strong covalent $Th^{3+}$–$Th^{3+}$ bond inside an $I_h(7)$-$C_{80}$ fullerene cage

Jiaxin Zhuang[1,8], Roser Morales-Martínez[2,8], Jiangwei Zhang [3,8], Yaofeng Wang[1,8], Yang-Rong Yao[4], Cuiying Pei[5], Antonio Rodríguez-Fortea [2], Shuao Wang [6], Luis Echegoyen[4], Coen de Graaf [2,7], Josep M. Poblet[2✉] & Ning Chen [1✉]

The nature of the actinide-actinide bonds is of fundamental importance to understand the electronic structure of the 5$f$ elements. It has attracted considerable theoretical attention, but little is known experimentally as the synthesis of these chemical bonds remains extremely challenging. Herein, we report a strong covalent Th-Th bond formed between two rarely accessible $Th^{3+}$ ions, stabilized inside a fullerene cage nanocontainer as $Th_2@I_h(7)$-$C_{80}$. This compound is synthesized using the arc-discharge method and fully characterized using several techniques. The single-crystal X-Ray diffraction analysis determines that the two Th atoms are separated by 3.816 Å. Both experimental and quantum-chemical results show that the two Th atoms have formal charges of +3 and confirm the presence of a strong covalent Th-Th bond inside $I_h(7)$-$C_{80}$. Moreover, density functional theory and ab initio multireference calculations suggest that the overlap between the 7$s$/6$d$ hybrid thorium orbitals is so large that the bond still exists at Th-Th separations larger than 6 Å. This work demonstrates the authenticity of covalent actinide metal-metal bonds in a stable compound and deepens our fundamental understanding of $f$ element metal bonds.

[1] College of Chemistry, Chemical Engineering and Materials Science, and State Key Laboratory of Radiation Medicine and Protection, Soochow University, Suzhou, Jiangsu, China. [2] Departament de Química Física i Inorgànica, Universitat Rovira i Virgili, Tarragona, Spain. [3] State Key Laboratory of Catalysis, Dalian Institute of Chemical Physics, Chinese Academy of Sciences (CAS), Dalian, China. [4] Department of Chemistry, University of Texas at El Paso, El Paso, Texas, USA. [5] Center for High Pressure Science and Technology Advanced Research, Pudong District, Shanghai, China. [6] State Key Laboratory of Radiation Medicine and Protection, School for Radiological and Interdisciplinary Sciences (RAD-X), and Collaborative Innovation Center of Radiation Medicine of Jiangsu Higher Education Institutions, Soochow University, Suzhou, China. [7] ICREA, Pg Lluis Companys 23, Barcelona, Catalonia, Spain. [8]These authors contributed equally: Jiaxin Zhuang, Roser Morales-Martínez, Jiangwei Zhang, Yaofeng Wang. ✉email: chenning@suda.edu.cn; josepmaria.poblet@urv.cat

Understanding the nature of chemical bonds is at the center of chemistry and is fundamental for the prediction of reactivity and consequent synthetic work. Many chemical compounds contain metal–metal bonds that are responsible for their structural and functional properties. Bonding between main group and $d$ transition group elements have been intensively studied for many decades and significant advances have been achieved on the synthesis of $f$–$d$ metal–metal bonds in recent studies[1–3]. In contrast, experimental studies of bonding between $f$-block metals is notably lagging behind[4]. Exploring the electronic structure of the $5f$ elements is crucial given their potential applications in different fields, such as in nuclear fuel recycling. Theory predicts that the $U_2$ molecule should be stable in the gas phase[5,6], and experimental evidence for its existence was reported in 1974 using mass spectrometry[7]. However, a long standing Holy Grail in inorganic chemistry is the synthesis and characterization of compounds that contain An–An bonds, and very little has been reported in this field. Most of the work concerning An–An bonds has been of a theoretical nature[5,6,8–11]. On the experimental side, Souter et al. reported the formation of $U_2H_2$ and $U_2H_4$ via laser ablation of U atoms in the presence of $H_2$, stabilized in a solid argon matrix[12]. However, characterization was limited to infrared spectroscopy supplemented with Density Functional Theory (DFT) calculations. The authors stated in the article abstract that: 'The molecules U($\mu$-$H_2$)U and $U_2H_4$ represent the first examples of an actinide-actinide bond'. To our knowledge, there are only two other reports that claim the formation of 'weak' U–U bonds, one in the $[U_2F_{12}]^{2-}$ anion in $Sr[U_2F_{12}]$[13] and our recent report of $U_2@I_h(7)$-$C_{80}$[14,15]. A molecular compound containing an An–An bond still remains highly sought after, as the synthesis of such a compound is extremely challenging, due to the fact that the actinide metals prefer to coordinate to the main group elements in the ligands instead of to other actinides. Recently, significant progress has been made with low oxidation state actinide chemistry[16–18]. Formation of a direct An–An bond between low oxidation state actinides has been theoretically proposed[11]. However, these low oxidation state actinides, including Th(III) and U(II), are still hardly accessible and only a very few examples have been reported[17,19].

Fullerenes are known to confine metal ions within a short distance and to stabilize metal ions with usual oxidation states, which provides an ideal environment for the formation of metal–metal bonds[20–25]. The possible existence of an actinide metal–metal bond inside fullerene cages has also been intensively debated by theoreticians in recent years[26,27]. Here, we report the synthesis and characterizations of a dimetallic thorium endohedral metallofullerene (EMF), $Th_2@I_h(7)$-$C_{80}$, containing a strong covalent thorium–thorium bond between two rarely accessible $Th^{3+}$ ions. This molecule represents an authentic example of an actinide metal–metal bond. Moreover, our theoretical studies show that the overlap between $7s/6d$ hybrid thorium orbitals is so large that the bond still exists at Th–Th separations larger than 6 Å.

## Results

**Synthesis and isolation of Th₂@C₈₀.** $Th_2@C_{80}$ was synthesized by a modified Krätschmer–Huffman arc discharge method. Graphite rods, packed with $ThO_2$ and graphite powder (molar ratio of Th/C = 1:24), were vaporized in the arcing chamber under a 200 Torr He atmosphere. The resulting soot was then extracted with $CS_2$ for 12 h. A multistage HPLC procedure was employed to isolate and purify $Th_2@C_{80}$. After a four-stage HPLC separation protocol, the purified $Th_2@C_{80}$ was obtained (Supplementary Fig. 2). The purity of the isolated $Th_2@C_{80}$ was confirmed by the observation of a single peak by HPLC and by high-resolution matrix-assisted laser desorption-ionization time-of-flight positive-ion-mode mass spectrometry (MALDI-TOF/MS). The mass spectrum shows a prominent molecular ion peak with a mass-to-charge ratio of $m/z$ = 1424.076 (Supplementary Fig. 1), corresponding to the empirical formula of $[Th_2C_{80}]^+$ and the experimental isotopic distribution agrees well with the theoretical prediction.

**Molecular structure of Th₂@$I_h$(7)-C₈₀·[Ni$^{II}$-OEP].** The molecular structure of $Th_2@C_{80}$ was determined by single-crystal X-ray diffraction analysis. $Th_2@C_{80}$ was co-crystallized with $Ni^{II}$-OEP (OEP = 2, 3, 7, 8, 12, 13, 17, 18-octaethylporphyrin dianion) by slow diffusion of a benzene solution of $Ni^{II}$-OEP into a $CS_2$ solution of $Th_2@C_{80}$. The structure was resolved and refined in the $P2_1/c$ (No. 14) space group. Fig. 1a shows $Th_2@I_h(7)$-$C_{80}$ and its relationship to the co-crystallized $Ni^{II}$(OEP) molecule. The crystallographic data indicate that the $I_h(7)$-$C_{80}$ cage is fully ordered. Inside the fullerene cage, two major Th positions (Th1 and Th2) are highly ordered and have dominant occupancies of 0.783(17) and 0.748(2), respectively, with the occupancies of minor sites

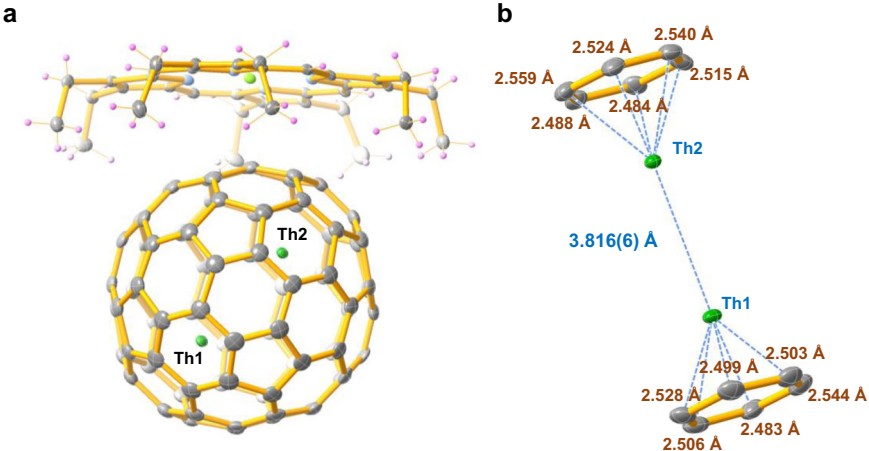

**Fig. 1 ORTEP drawing of Th₂@$I_h$(7)-C₈₀·[Ni$^{II}$(OEP)] with 20% thermal ellipsoids. a** $Th_2@I_h(7)$-C80·[NiII(OEP)] structure showing the relationship between the fullerene cage and the [Ni$^{II}$-OEP] ligands. The two Th1/Th2 sites have occupancies of 0.783(17) and 0.748(2), respectively. Other minor Th sites (Supplementary Fig. 3) and the solvent molecules are omitted here for clarity. **b** Fragment view showing the interaction of the major Th1–Th2 cluster with the closest aromatic ring fragments of the cage.

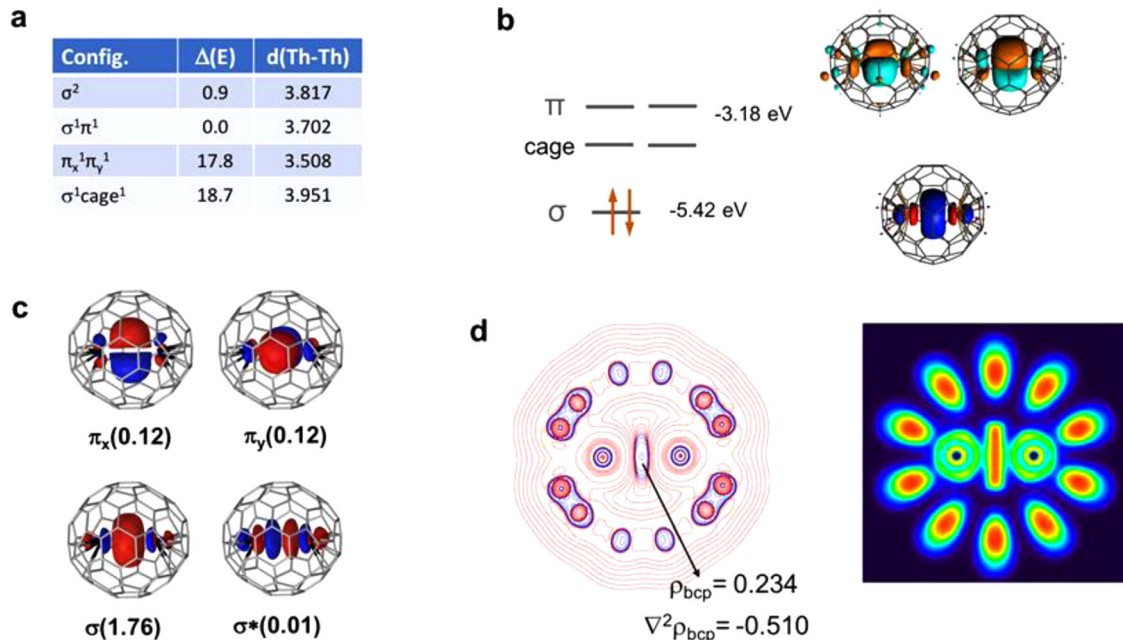

**Fig. 2 Structural and electronic properties for the dithorium endohedral fullerene Th$_2$@$I_h$(7)-C$_{80}$. a** Th–Th distances (in Å) and relative energies (in kcal mol$^{-1}$) computed for several electronic states at the DFT (PBE0) level. **b** Scheme containing selected DFT PBE0 frontier molecular orbitals. **c** Representation of CASSCF orbitals for the singlet ground state; the electron population for each orbital (in parenthesis) leads to an effective Th–Th bond order (EBO) of 0.99. **d** Representation of the Laplacian of the electron density distribution and electron localization function (ELF) in a plane containing the two Th atoms. Blue and red contours in the Laplacian of $\rho$ represent negative and positive values, respectively.

ranging from 0.0028(7) to 0.1299(14) (Supplementary Table 1). Most of the reported lanthanide dimetallic endohedral fullerenes as well as the recently reported U$_2$@$I_h$(7)-C$_{80}$ commonly display severe disorder of the metals inside the highly symmetric $I_h$(7)-C$_{80}$[14,28]. The highly ordered Th positions inside this cage are very unusual for pristine dimetallic endohedral fullerenes, suggesting a hindered metal motion, possibly caused by the bonding interactions between the Th ions and the fullerene cage as well as between the two Th ions. The ordered Th positions allow us to accurately determine the distance between major Th1 and Th2 as 3.816(6) Å. This distance is slightly longer than the U–U and La–La distances reported for U$_2$@$I_h$(7)-C$_{80}$[14] and La$_2$@$I_h$-C$_{80}$(C$_7$H$_7$)[22] but is comparable to the Ce–Ce bond distance in Ce$_2$@$I_h$-C$_{80}$[28] and even shorter than the metal–metal bond distance observed in La$_2$@$I_h$-C$_{80}$[29], Dy$_2$@$I_h$-C$_{80}$(CH$_2$Ph)[21], and Tb$_2$@C$_{79}$N[20] (Supplementary Table 3). Th1 and Th2 reside under corresponding hexagons and the distances between the major thorium sites and the cage carbons all lie within the range of 2.483 (9)–2.559(9) Å, with Th1 and Th2 at distances to their corresponding hexagon centroids of 2.048 and 2.066 Å, respectively. (See Fig. 1b, Supplementary Table 2, and Supplementary Fig. 4). It is noteworthy that these Th-(ring centroid) distances are shorter than those for previously reported Th$^{3+}$ complexes: 2.607, 2.52, 2.53, and 2.551 Å for (C$_5$Me$_5$)$_3$Th[17], [C$_5$H$_3$(SiMe$_3$)$_2$]$_3$Th[30], [C$_5$H$_3$(SiMe$_2$$^t$Bu)$_2$]$_3$Th[30], and (C$_5$Me$_4$H)$_3$Th[31], respectively, indicating a strong interaction between the Th$_2$ cluster and the fullerene cage.

**Electronic structure and bonding in Th$_2$@$I_h$(7)-C$_{80}$.** DFT and ab initio calculations were performed to determine the electronic structure of the Th$_2$@$I_h$(7)-C$_{80}$ EMF and to characterize the nature of the Th–Th interaction. Since no more than 6e are transferred between the guest and the host[32,33], the oxidation state of the Th atoms in Th$_2$@$I_h$(7)-C$_{80}$ should be +3, which is a rare situation given the tendency of Th atoms to formally transfer all four valence electrons to the groups with which they coordinate[17,34–36]. Accordingly, two electrons are available to form a Th–Th bond. At the DFT level, we have computed several structures corresponding to the electronic configurations $\sigma^2$, $\sigma^1\pi^1$, and $\pi^2$. When the two valence electrons are localized in the $\sigma$ orbital, the Th···Th separation is computed to be 3.817 Å using the hybrid functional PBE0, a value that matches the experimental distance of 3.816 Å. The nature of the HOMO, a bonding $\sigma$ Th–Th orbital (Fig. 2b), suggests that despite the long Th–Th distance, the two Th atoms are linked by a direct interaction. Although the LUMO is delocalized over the carbon cage, the lowest triplet corresponds to the promotion of one electron from the HOMO to a bonding orbital with a $\pi$ ($6d_{xz}$, $6d_{yz}$) nature and, interestingly, the Th–Th distance shortens by 0.114 Å to 3.703 Å. If both $\sigma$ electrons are transferred to the $\pi$ orbitals, the shortening is still larger, 0.309 Å. At the PBE0 level, the energy of the singlet ($\sigma^2$) and triplet ($\sigma^1\pi^1$) are almost degenerate with the triplet being about 1 kcal mol$^{-1}$ lower in energy (Fig. 2a). Using the GGA PBE functional, the energy difference reaches 4 kcal mol$^{-1}$ (Supplementary Table 5).

To properly characterize the nature of the ground state, we have performed CASPT2 calculations using the singlet DFT-optimized structure. As shown in Table 1, the ab initio ground state is a singlet with two electrons mainly localized in a $\sigma$ ($7s$, $6d_z$) orbital. The $\pi_x$ and $\pi_y$ bonding orbitals make a significant contribution (12%) to the wave function, whereas the antibonding $\sigma^*$ orbital has an almost null contribution (Fig. 2c). Thus, the Th–Th bond can be seen as a $\sigma$ bond with some contribution of bonding $\pi$ orbitals to give an effective bond order of 0.99. When one electron is promoted from the $\sigma$ orbital to one of the $\pi$ orbitals the relative energy increases to 11.4 kcal mol$^{-1}$ (9.5 kcal mol$^{-1}$ with the triplet DFT structure). The discrepancy between DFT and CASPT2 results arises from the nature of the two electronic states. Whereas the triplet state is mono configurational, the singlet ground state has an important multiconfigurational nature that cannot be correctly described

**Table 1 CASPT2 energies and effective bond orders for Th$_2$@$I_h$(7)-C$_{80}$ and Th$_2$@$D_2$(821)-C$_{104}$ EMFs [a].**

| Cage | d$_{Th-Th}$ | Config. | Spin | ΔE | EBO [b] |
|---|---|---|---|---|---|
| $I_h$(7)-C$_{80}$ | 3.817 | σ$^2$ | singlet | 0.0 | 0.99 |
| | | σ$^1$π$^1$ | triplet | 11.4 | 1.00 |
| | | π$^2$ | triplet | 37.3 | 0.91 |
| | | σ$^1$cage$^1$ | triplet | 45.4 | 0.50 |
| | | π$^1$ π$^{*1}$ | triplet | 86.6 | 0.00 |
| | 3.702 | σ$^1$π$^1$ | triplet[c] | 9.5 | 1.00 |
| $D_2$(821)-C$_{104}$ | 6.020 | σ$^2$ [d] | singlet | 0.0 | 0.76 |
| | | σ$^1$cage$^1$ | singlet | 13.4 | 0.50 |
| | | σ$^1$π$^1$ [e] | singlet | 18.8 | 0.00 |
| | | π$^1$cage$^1$ | singlet | 36.0 | 0.50 |

[a] CASSCF and CASPT2 calculations were performed at the ground state PBE0 geometries. All energies are given in kcal mol$^{-1}$.
[b] Effective Th-Th bond orders computed from CASSCF active MO occupations.
[c] Energy of the triplet computed using the DFT geometry of the triplet.
[d] MO occupations σ(1.73) σ*(0.24).
[e] MO occupations σ(0.70) σ*(0.30) π(0.30) π*(0.70).

at the DFT level. In all DFT and CASPT2 calculations, only two electrons reside in molecular orbitals formed mainly by valence-shell Th orbitals, which confirms the oxidation state of +3 for the Th ions in the lowest energy states.

Both DFT and CASPT2 calculations suggest that the Th–Th interaction inside the icosahedral C$_{80}$ is covalent and clearly cannot be considered as a magnetic interaction. The covalent bond arises from the nature of the 7$s$ and 6$d$ actinide orbitals that extend their overlap over very long distances, as shown in Supplementary Fig. 6. Because the overlap between Th orbitals can extend over distances larger than 6 Å, we have analyzed the thorium–thorium interaction inside cage $D_2$(821)-C$_{104}$, which can be visualized as two hemispheres of $I_h$(7)-C$_{80}$ with 24 additional carbon atoms located in the center of the cage (see Supplementary Fig. 7a). As for C$_{80}$, the HOMO for the lowest singlet of Th$_2$@$D_2$(821)-C$_{104}$ corresponds to a σ bonding orbital localized on Th atoms (Supplementary Fig. 7c), the Th–Th separation being significantly longer, 6.020 Å, due to the larger size of the fullerene. CASPT2 calculations were also performed for Th$_2$@$D_2$(821)-C$_{104}$, confirming the singlet nature of the ground state. The increase of the Th–Th distance with respect to C$_{80}$ necessarily reduces the covalent interaction between the two actinides. A direct consequence is the presence of a relatively small energy gap between the σ(7$s$/6$d_{z2}$) and σ*(7$s$/6$d_{z2}$) orbitals. A partial population of the σ* orbital is observed in the CASSCF ground-state wave function, which leads to an effective bond order of 0.76 for the Th–Th bond, still very high considering the very long separation between the two Th atoms.

Finally, the Th–Th interaction was also characterized using Baders's Quantum Theory of Atoms in Molecules (QTAIM). This theory uses several descriptors to characterize bonds through the topology of the electron density[37]. Bader postulated that the presence of a bond critical point (bcp) between two atoms is a necessary and sufficient condition for the atoms to be bonded. Such a bcp was found that links the two Th atoms in Th$_2$@$I_h$(7)-C$_{80}$. A representation of the Laplacian of the electron density ($\nabla^2 \rho_{bcp}$) distribution for the singlet ground state is given in Fig. 2d, which exhibits a non-negligible negative region in the internuclear Th–Th region. A direct consequence is that the value of the Laplacian of the charge density at the bcp is negative. These results are remarkable since metal–metal bonds typically exhibit depletion of the electron density in the internuclear region, that is, positive values in the Laplacian of $\rho$. The ELF plot also shows that the electron density is highly localized in the internuclear Th–Th bonding region[38]. Indeed, when comparing the

topological descriptors for Th$_2$, U$_2$, Lu$_2$, and La$_2$ bonds inside C$_{80}$ and C$_{82}$ (Supplementary Table 6), we have found that the thorium–thorium bond exhibits the largest values even though its bond distance is the longest. Therefore, the thorium–thorium interaction is, up to now, the strongest one among the known actinide–actinide or lanthanide–lanthanide interactions.

As one cannot dissociate the Th$_2$ unit inside the cage, it is not easy to give an accurate value for the Th–Th bond energy in Th$_2$@C$_{80}$. A rough estimate can be obtained from the relative energy of the triplet corresponding to the σ$^1$σ$^{*1}$ configuration, as in this electronic configuration the Th–Th bond is effectively broken. However, the antibonding orbital is very high in energy and only the energies corresponding to the configurations σ$^1$π$^1$ and σ$^1$cage$^1$ could be calculated. The latter triplet is about 2 eV higher in energy than the singlet ground state at CASPT2 level (Table 1). An alternative procedure is to compare the difference in the encapsulation energies of two La atoms and two Th atoms in C$_{80}$, which is 1.92 eV at the PBE0 level (1.7 eV at PBE). Assuming that the metal–cage interaction is approximately the same in both cases and that no bond is formed in the La$_2$ case, the difference in encapsulation energy arises from the metal–metal bond in the dithorium EMF. Both procedures are rather approximate and have limitations, but it does not seem unreasonable to state that Th–Th bond energy is greater than 40 kcal mol$^{-1}$. It is worth mentioning that for cage $D_2$(821)-C$_{104}$, the difference in the encapsulation energies between dithorium and dilanthanum EMFs is still 0.45 eV at the PBE0 level and 0.25 eV at PBE (Supplementary Table 7). As already mentioned above, this strong metal–metal bond interaction is a result of a high overlap between 7$s$6$d$ hybrid orbitals which extend over long distances (Supplementary Fig. 6). We are persuaded that this property could also be used in more traditional chemistry to obtain long-distance interactions between actinides, for example, through the design of appropriate ligands in organometallic chemistry. It is worth mentioning that uranium–uranium interactions are not that strong over long distances, as they occur through the more contracted 5$f$ orbitals, their overlap decreasing much more rapidly with U–U separation. Actually, according to CASPT2 calculations, two U atoms within $I_h$-C$_{80}$ (d$_{U-U}$ = 3.812 Å) show magnetic coupling with an effective bond order of 0.09. A detailed analysis of under what conditions two U atoms can exhibit a covalent interaction will be reported elsewhere.

**Spectroscopic properties of Th$_2$@C$_{80}$.** Purified Th$_2$@$I_h$(7)-C$_{80}$ was characterized with UV–Vis–NIR and Raman spectroscopies as shown in Supplementary Fig. 5 and Fig. 3. The UV–Vis spectrum of Th$_2$@ C$_{80}$ shows three weak absorptions at 447, 488, and 683 nm, similar to typical $I_h$(7)-C$_{80}$ cage based endofullerenes[39]. Notably, a single vibration peak at 152 cm$^{-1}$ can be observed in the low energy Raman spectrum and is assigned to a metal-to-cage vibration based on previous studies[40]. This frequency is very close to 148 cm$^{-1}$ for Th@$C_{3v}$(8)-C$_{82}$[41], 153 cm$^{-1}$ for Th@$C_1$(11)-C$_{86}$[42], and 155 cm$^{-1}$ for Th@$C_1$(28324)-C$_{80}$[43], indicating similarly strong thorium–cage interactions. Moreover, the peaks at 220, 416 and 475 cm$^{-1}$ also observed in the Raman spectra of U$_2$@$I_h$(7)-C$_{80}$[14] and U$_2$C@$I_h$(7)-C$_{80}$[44], can be assigned to the $I_h$(7)-C$_{80}$ cage vibrational modes, consistent with the crystallographic analysis. The Raman peaks obtained at the DFT PBE level coincide well with the experimental ones (see Fig. 3). The computed frequencies for the optimized structure show six lower frequencies that correspond to frustrated translational and rotational modes against the cage (at 40, 47, 57, 67, and 165 cm$^{-1}$), and the internal Th–Th stretching (at 138 cm$^{-1}$), which is coupled with a cage mode (Fig. 3), accounting for Th–cage and Th–Th interactions. Smaller contributions of the

symmetric Th–Th stretching vibrations are found coupled with many other cage vibrations and appear at higher frequencies.

The $^{13}C$ NMR spectrum of $Th_2@I_h(7)$-$C_{80}$ characterized at 298 K show four signals in the range 125–145 ppm as shown in Supplementary Fig. 9. Though there are two signals identified as

contamination caused by benzene impurities, signals at 144.05 and 129.27 ppm could be assigned to characteristic signals for a $C_{80}$ cage with $I_h$ symmetry. In particular, there is no paramagnetic broadening effect found in the $^{13}C$ NMR spectrum of $Th_2@I_h(7)$-$C_{80}$, further proving that $Th_2@I_h(7)$-$C_{80}$ displays diamagnetic features and two electrons are available to form a Th–Th bond, agreeing well with the above computational results.

To determine the valence state of thorium inside the $I_h(7)$-$C_{80}$ cage, the X-ray absorption near edge structure (XANES) spectroscopy of $Th_2@I_h(7)$-$C_{80}$ was conducted along with the $ThO_2$ ($Th^{4+}$) as the reference species. The oxidation state of the metal in a statistical nature can be analyzed from the XANES region based on the edge position and the inflection point of the edge absorption. In terms of the edge position, the Th $L_3$-edge spectrum of $Th_2@C_{80}$ is shifted to lower energy side compared to that of the $ThO_2$ as shown in Fig. 4a. In addition, the maximum peak position in the first derivative of normalized $\mu(E)$ spectra is defined as $E_0$, namely the inflection point energies. In particular, the $E_0$ found for $Th_2@C_{80}$, 16306.7 eV, is shifted to lower energy by 4.7 eV compared with that of $ThO_2$, 16311.4 eV (see Fig. 4a inset and Supplementary Fig. 8), indicating that the two Th atoms encapsulated in the $I_h(7)$-$C_{80}$ cage have a lower charge state and agrees well with the theoretical predictions that the two Th ions exist in +3 oxidation state.

Moreover, the extended X-ray absorption fine structure (EXAFS) spectroscopy further provides information of the coordination environment of Th atoms. The radial distance space spectrum $\chi(R)$, obtained by the Fourier transform of the Th $L_3$-edge EXAFS oscillation, $\chi(k)$ of $Th_2@I_h(7)$-$C_{80}$ (k range 2.54–10.42 $Å^{-1}$), is presented as the radial distribution function including the phase shift (+0.41 Å). This shift is determined by the difference between the first shell Th–C distance from single crystal structure (2.48 Å) and the first scattering path (2.07 Å). As

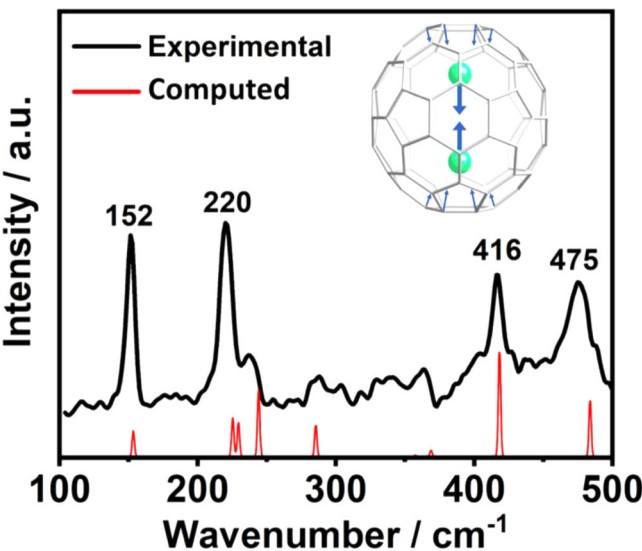

**Fig. 3 Observed and simulated Raman spectra of $Th_2@I_h(7)$-$C_{80}$.** The low-energy Raman spectrum of $Th_2@C_{80}$ with 633 nm excitation is shown in black. Computed spectrum (in red) fits well with the experimental one if all the frequencies are shifted by 15 $cm^{-1}$ to higher values. The symmetric stretching mode of $Th_2$ (at 152 $cm^{-1}$) is active in Raman and is coupled with a cage–Th vibration. The associated normal mode is also represented.

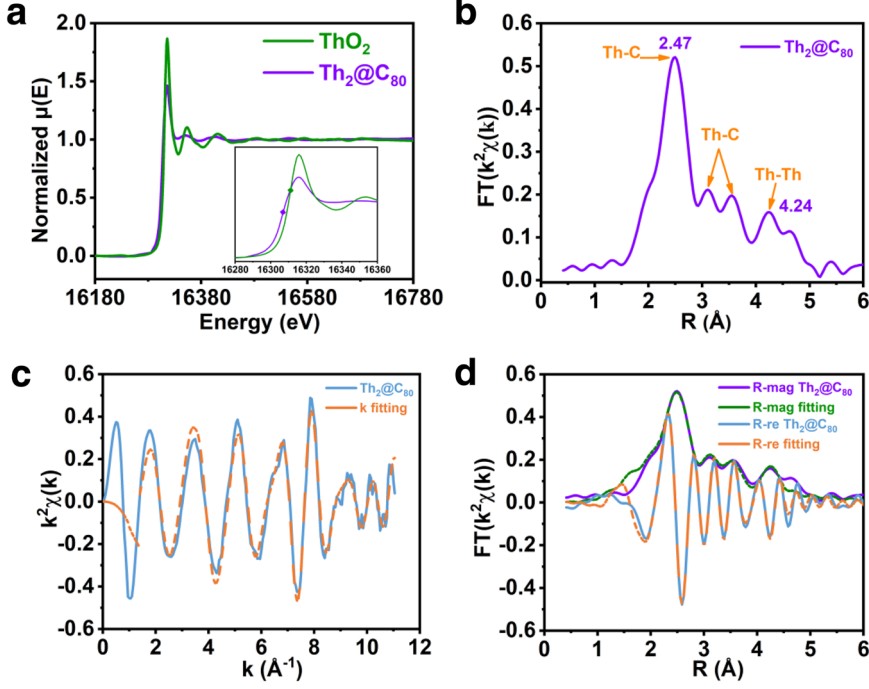

**Fig. 4 X-ray absorption analyses of $Th_2@I_h(7)$-$C_{80}$.** **a** $L_3^-$edge normalized $\mu(E)$ spectra of Th in $Th_2@I_h(7)$-$C_{80}$ (purple) and $ThO_2$ (green). The inset shows an enlarged spectral range (16280–16360 eV) and the diamonds indicate the absorption edges $E_0$. **b** Radial distribution function obtained by Fourier transform of $k^2$-weighted Th $L_3$-edge EXAFS spectrum of $Th_2@I_h(7)$-$C_{80}$ measured at room temperature. **c** $\chi(k)$ space (blue) and fitting curve (orange) of $Th_2@I_h(7)$-$C_{80}$. **d** Radial distribution function of $k^2$-weighted Th $L_3$-edge EXAFS spectrum and fitting curves of $Th_2@I_h(7)$-$C_{80}$, including the magnitude (R-mag in purple) and real part (R-re in blue) of the Fourier transform of the EXAFS data of $Th_2@I_h(7)$-$C_{80}$ along with their fitting curves (R-mag fitting in green and R-re fitting in orange).

shown in Fig. 4b, the predominant peak located around 2.47 Å can be assigned as the scattering path of Th–C interaction from the first coordination shell in $Th_2@I_h(7)$-$C_{80}$, which is consistent with the ferrocene like Th–C bonds observed in single crystal structure shown in Fig. 1b. Besides the relative strong first Th–C interactions, the observation of the second and third Th–C interaction suggests that Th ions may also have weaker but non-negligible interaction with carbon atoms in wider region. As shown in Supplementary Figs. 12 and 13, the Th–C distances of 2.90 Å and 3.35 Å (average values) can be found in the crystal structure of $Th_2@C_{80}$, which correspond to the second and third Th–C interaction peaks at 3.11 and 3.54 Å, respectively. Note that the differences between Th–C distances found in SCXRD and EXAFS might be ascribed to different methodologies as well as the different measure temperatures, as the single crystal X-ray data were collected at 113 K and the XAS experiment was carried out at room temperature (ca. 298 K).

Interestingly, the prominent peak located around 4.24 Å can be attributed to the scattering path of Th–Th bond in fourth coordination shell in $Th_2@I_h(7)$-$C_{80}$. Noteworthy that due to the obvious attenuation in higher $\chi(R)$, the strength of Th–Th bond with long distance in fourth coordination is not as strong as the typical metallic bond like Fe–Fe bond in first coordination of Fe foil[45]. However, it still can be recognized since there is no scattering path signal of Th–C bond at this distance. Note that the scattering path of Th–Th interaction is a little longer than the bond distance (3.816 Å) observed from single crystal structure due to the phase shift.

Finally, quantitative $\chi(R)$ space spectra fitting with the $k$ range from 0 to 11.06 Å$^{-1}$ were also performed to validate local atomic structure and further obtain the bonding type and coordination numbers of Th in $Th_2@I_h(7)$-$C_{80}$ (Fig. 4c). The single crystal structure of $Th_2@I_h(7)$-$C_{80}$ shows that there is no bond existing under 2 Å, thus all the scattering path signals from 2 to 6 Å have been taken into the fitting consideration. The $\chi(R)$ space spectrum of $Th_2@I_h(7)$-$C_{80}$ displays Th–C bond with coordination number (CN) approaching 6.0 at 2.477 Å, 6.0 at 3.100 Å, and 8.0 at 3.563 Å, respectively (Supplementary Table 8), from first to third coordination shell which can be connected to Fig. 1b, Supplementary Figs. 12 and 13, respectively. Moreover, the obvious Th–Th metallic bond with coordination number (CN) approaching 1.0 at 4.24 Å from fourth coordination shell can be also observed. The good fitting results of $\chi(k)$, $\chi(R)$, and real $\chi(R)$ space spectra, as shown in Fig. 4d, with reasonable R-factor (0.0218%, Supplementary Table 8) and the obtained fitting parameters quantitatively confirms the coexistence of Th–C and Th–Th bonds. Thus, the two $Th^{3+}$ ions confined in $C_{80}$ with Th–Th metallic bonding formation and the ferrocene like Th–C bonds can be confirmed by the combination of single crystal X-ray diffraction and XAFS characterization results, which agree well with the above discussed theoretical calculation results.

## Discussion

In summary, we report the formation and characterization of a strong bond between actinides with low oxidation states inside a fullerene cage, namely, $Th_2@I_h(7)$-$C_{80}$. This compound was synthesized by the arc-discharge method. The single-crystal X-ray diffraction unambiguously determined that the two encapsulated Th atoms are separated by 3.816 Å inside an $I_h(7)$-$C_{80}$. The overall agreement between the crystallographic, XAS, and quantum-computational results conclusively shows that the two encapsulated Th ions have formal charges of +3 and confirm the presence of a strong covalent Th–Th bond inside $I_h(7)$-$C_{80}$, estimated to be ≥40 kcal mol$^{-1}$. The computational studies reveal that the overlap between $7s/6d$ hybrid thorium orbitals is so large

that the bond still prevails at Th–Th separations larger than 6 Å, suggesting that this thorium–thorium bond is, up to now, the strongest one among the known actinide–actinide or lanthanide–lanthanide interactions.

This study presents an isolated compound with a strong covalent actinide metal-metal bond, paving the way for future experimental studies of these fundamentally important yet far from fully understood bonding motifs. Moreover, this study proves that the unique bonding environment inside fullerene cages make actinide endohedral fullerenes unique prototype molecules to study actinide–actinide bonding in detail. Because encapsulation always restricts the number of electrons that are transferred from the guest to the fullerene at ≤6, diactinide endohedral fullerenes always have two or more electrons that are available to form single or multiple An–An bonds. Such bonds remain inaccessible by conventional synthetic procedures. These metal–metal bonds may help in the formation mechanisms of dimetallic endohedral fullerenes and it is conceivable that stronger An–An bonds could be formed inside smaller fullerene cages. Further studies are under way to better understand the multiple factors involved in the formation of encapsulated actinide–actinide clusters and the stabilization of the corresponding endohedral fullerenes.

## Methods

**Synthesis, separation, and purification of $Th_2@C_{80}$.** The carbon soot containing thorium EMFs was synthesized by the direct-current arc discharge method. The graphite rods, packed with $ThO_2$ and graphite powder (molar ratio of Th/C = 1:24), were annealed in a tube furnace at 1000 °C for 20 h under an Ar atmosphere and then vaporized in the arcing chamber under 200 Torr He atmosphere. In total 1.87 g of graphite powder and 1.73 g of $ThO_2$ were packed in each rod. The resulting soot was refluxed in $CS_2$ under an argon atmosphere for 12 h. On average ca. 210 mg of crude fullerene mixture per rod was obtained. In total, 100 carbon rods were vaporized in this work. The separation and purification of $Th_2@C_{80}$ was achieved by a multistage HPLC procedure (Supplementary Fig. 2). Multiple HPLC columns, including Buckyprep M (25 × 250 mm, Cosmosil, Nacalai Tesque Inc.), Buckyprep-D (10 × 250 mm, Cosmosil, Nacalai Tesque, Japan), and Buckyprep (10 × 250 mm, Cosmosil, Nacalai Tesque, Japan), were utilized in this procedure. Toluene was used as the mobile phase and the UV detector was adjusted to 310 nm for fullerene detection. The HPLC traces and corresponding MALDI-TOF spectra for the isolated fractions are shown in Supplementary Fig. 2. In total, ca. 1 mg of highly purified $Th_2@C_{80}$ was obtained for characterization.

**X-ray crystallographic study.** The black block crystals of $Th_2@I_h(7)$-$C_{80}$·[$Ni^{II}$·(OEP)] were obtained by slow diffusion of a carbon disulfide solution of $Th_2@C_{80}$ into a benzene solution of [$Ni^{II}$·(OEP)]. X-ray data were collected at 113 K using a diffractometer (Bruker D8 Venture) equipped with a CCD detector. The multi-scan method was used for absorption correction. The structure was solved using direct methods[46] and refined on $F^2$ using full-matrix least-squares using SHELXL2015[47]. Hydrogen atoms were inserted at calculated positions and constrained with isotropic thermal parameters.

Crystal data for $Th_2@I_h(7)$-$C_{80}$·[$Ni^{II}$(OEP)]·1.5$C_6H_6$·$CS_2$: Mr = 2209.63, 0.2 × 0.2 × 0.2 mm, monoclinic, $P2_1/c$ (No. 14), $a$ = 17.7202(14) Å, $b$ = 17.0187(13) Å, $c$ = 26.800(2) Å, $\alpha$ = 90°, $\beta$ = 106.811(2)°, $\gamma$ = 90°, $V$ = 7736.7(10) Å$^3$, $Z$ = 4, $\rho_{calcd}$ = 1.897 g cm$^{-3}$, $\mu$(Cu Kα) = 13.585 mm$^{-1}$, $\theta$ = 3.12 – 68.26, $T$ = 113(2) K, $R_1$ = 0.0591, $wR_2$ = 0.1682 for all data; $R_1$ = 0.0587, $wR_2$ = 0.1678 for 13898 reflections (I > 2.0σ(I)) with 1284 parameters. Goodness-of-fit indicator 1.048. Maximum residual electron density 3.590 e Å$^{-3}$.

**XAFS measurements.** The X-ray absorption structure spectroscopy (Th L$_3^-$edge) were collected at BL11B in Shanghai Synchrotron Radiation Facility (SSRF). The samples were filled into the hole with 1.5 mm diameter on PTFE films (film thickness ca. 0.2 mm) for test. Presented data were produced as an average of three consecutive scans for $Th_2@C_{80}$ while two consecutive scans for $ThO_2$.

**Computational methods.** DFT calculations for $Th_2@C_{2n}$ EMFs were carried out with the ADF 2017 package[48] using PBE and PBE0 exchange-correlation functionals in combination with Slater TZP basis sets to describe the valence electrons of Th and C[49,50]. Frozen cores were described by means of single Slater functions, consisting of the 1 s shell for C and the 1 s to 5d shells for Th. Scalar relativistic corrections were included by means of the ZORA formalism. Dispersion corrections by Grimme were also included[51]. Relatives energies were computed with GGA PBE and BP86 functionals reoptimizing the structures with each functional.

CASPT2 calculations were performed with OpenMolcas[52]. The active space contains six orbitals and two electrons in all cases. Test calculations with larger

active spaces do not alter the results; Th-carbon bonding and anti-bonding orbitals enter the active space with occupation numbers close to 2 and 0, respectively. The computational costs were reduced by imposing the symmetry restrictions of the $D_{2h}$ group in the case of $Th_2@C_{80}$ and $D_2$ for $Th_2@C_{104}$. Furthermore, we have used the Cholesky decomposition of the two-electron integrals with a $10^{-4}$ threshold. Scalar relativistic effects were taken into account with the Douglas–Kroll–Hess Hamiltonian[53]. We have used the standard IPEA = 0.25 zeroth-order Hamiltonian in the CASPT2 calculations and applied an imaginary level shift of 0.15 eV to avoid intruder states. CASPT2 calculations were performed using PBE0 geometries obtained under the symmetry constraints.

## Data availability

The X-ray crystallographic coordinates for the structure reported in this article are available free of charge from the Cambridge Crystallographic Data Centre (CCDC) under deposition number 1961861. The data can be obtained free of charge from The Cambridge Crystallographic Data Centre via www.ccdc.cam.ac.uk/data_request/cif. A data set collection of computational results is available in the ioChem-BD repository and can be accessed via https://doi.org/10.19061/iochem-bd-2-41 [54]. All other data supporting the findings of this study are available from the corresponding authors on request.

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

## Acknowledgements

We cordially thank beamline BL11B of the Shanghai Synchrotron Radiation Facility (SSRF) for XAS test and Dr. Duo Zhang (Soochow University) for the kind technique support. C.N. thanks the National Science Foundation China (NSFC 91961109, 51302178), the Natural Science Foundation of Jiangsu Province (BK20200041), and Priority Academic Program Development of Jiangsu Higher Education Institutions (PAPD). J.M.P. and C.d.G. thank the Spanish Ministry of Science (grants CTQ2017-87269-P and CTQ2017-83566-P), the Generalitat de Catalunya (grant 2017SGR629) and the URV for support. J.M.P. also thanks ICREA foundation for an ICREA ACADEMIA award. R.M.M. thanks Spanish Ministry of Science for a PhD fellowship. Z.J.W. thanks the National Natural Science Foundation of China (grant 21701168). L.E. thanks the US National Science Foundation (NSF) for generous support of this work under grant CHE-1801317, the Robert A. Welch Foundation is also gratefully acknowledged for an endowed chair to L.E. (grant AH-0033).

## Author contributions

N.C. conceived and designed the experiments. Z.J.X. and W.Y.F. synthesized and isolated all the compounds. R.M.M., A.R.F., C.d.G., and J.M.P. performed the computations and theoretical analyses. Y.Y.R and Z.J.X. performed the crystallographic analysis. Z.J.X. and P.C.Y. performed the XAS measurements and Z.J.W. performed the XAS analyses. N.C., J.M.P., Z.J.X., C.d.G., Z.J.W., R.M.M., A.R.F., L.E., and W.S.A. co-wrote the manuscript.

## Competing interests

The authors declare no competing interests.
