## [Peer Review File · Nature Communications]

Reviewers' comments:

Reviewer #1 (Remarks to the Author):

The report by Poblet and Chen and their respective co-workers detail the synthesis and characterization of C80-fullerene encaged Th(III)-Th(III) dimer. The experimental works appears to be carried out with care and the characterization of this species by X-ray crystallography, Raman spectroscopy, mass spectrometry and X-ray absorption analyses are consistent with their claims. Computational studies at the DFT and CASPT2 level of theory were employed to gain additional insights into the bonding of this interesting species. Overall it constitutes an interesting contribution to the field of actinide chemistry, which consequently deserves to be published. However, a very similar study on U2@Ih(7)-C80 was reported recently by the same leading authors in JACS in 2017 (ref. 14). Now they expanded this study to Th, and similar investigations concerning the 4f-elements (La, Ce, Dy, etc.) have also been detailed. The advancement herein compared to the previous ones is that Th apparently shows the strongest covalent interaction consistent with a Th-Th bond. Granted this is a novel and interesting finding, and I also admit that low-valent Th(III) species are rare, but they are not without precedent. The same holds true for Th(II) and U(II) species as correctly referenced by the authors. As a synthetic inorganic chemistry, I am also a little bothered about the significance of a reaction that yields 1 mg of a pure compound. Synthetically this is not a useful quantity and hampers any reactivity study which would be of significant interest to the community. In conclusion, while I would support publication of this contribution with respect to technical quality, I am not convinced that this paper carries enough importance and novelty to justify its publication in Nature Communications.

Some additional scientific requests:

- 1.) Can the singlet ground state be confirmed by magnetic susceptibility studies?
- 2.) How thermally stable is this species?

Reviewer #2 (Remarks to the Author):

The authors of the manuscript present the syntheses and characterisation of two Th atoms enclosed in a fullerene (Th₂@Ih(7)-C80). They also investigate the nature of the Th-Th bond by applying several modern theoretical approaches. Those analyses appear reasonable but surely a sound theoretician, expert in computations of actinides, should review them. As the authors mention, indeed there are a very few examples in the literature where An-An bond was stabilized and its properties were studied not only theoretically but also experimentally. However, considering that the first paper of the authors about U₂@C80 on basically the same topic was in JACS, I do wonder why this paper on Th should be in Nat. Commun.? The authors do not really present for example any novel spectroscopy method, which maybe provides a deeper inside on the electronic structure of the Th-Th bond. Until the XAS part I was actually still thinking that even if the paper should be in my opinion in a journal with a smaller impact factor due to the lack of novelty, it is still a very nice paper. The actinide community will find it from high interest. However, the X-ray absorption data and analyses are simply wrong. I strongly advise the authors to completely reconsider them. The paper can be only rejected from any journal in the current form. More specific comments follow:

- 1) The authors report that the Th XAS spectra in Fig. 4 a) are $\chi(E)$. this however look to me normalised $\mu(E)$ spectra. Usually, only the oscillating part of XAS spectra after the main absorption peak are called $\chi(E)/\chi(k)$.
- 2) The XAS spectrum of the Th₂@Ih(7)-C80 in Fig. 4 has several strange peaks/features at about 16200 eV (one feature) and at about 16600 eV (two features). Those look like artefacts, which are a problem for the analyses of the data.
- 3) The authors suggest that there is "pre-edge" at 16200 without giving any interpretation. This

however does not look at all like a spectral feature. The difference pointed out at 16600 is very likely also an artefact (see point 2).

4) Why were the XAS spectra modelled with Gaussian + arc-tangent function? The authors do not explicitly say this.

5) Both FT-EXAFS spectra in Fig. 4 a b look unusual and not correct. The FT-EXAFS of ThO₂ appears to have high noise level. The FT-EXAFS spectrum of Th₂@C₈₀ has one single peak, which looks almost like a Gaussian curve, which has an extremely smooth right slope. The FT-EXAFS spectra do not look like this even if the materials are highly disordered.

6) The authors claim "subtle peaks" between 2.3 -3 Å corresponding to Th-C bonds. There are no peaks visible at this region, which can be attributed to Th-C bonds.

7) I do not understand what is meant under "different distance states".

8) I have no opinion about the wavelengths analyses due to my limited experience with this type of analyses. But, the fitting of the EXAFS data is simply wrong. The authors suppose that they see Th-C scattering in the spectrum but do not take it into account in their modelling. They claim that their model is simplified without the Th-C bonds but do model the entire data, which should include this Th-C bonds.

9) It is not given the k range of the data, which was modelled. The structural parameters obtained from the EXAFS fit like coordination numbers, DW factors, S₀ and E₀ parameters are not given. The data in k range (not filtered) should be given together with the fit.

Reviewer #3 (Remarks to the Author):

This paper reported an important result: the synthesis and characterizations of a strong actinide-actinide bond, which is long sought after, but hardly accessible by conventional synthetic method so far. This compound was synthesized using the arc-discharge method and fully characterized using several techniques. The structure of Th₂@C₈₀ was unambiguously determined by single-crystal X-Ray diffraction analysis. A in-depth XAS analysis also confirms the presence of the strong covalent Th³⁺-Th³⁺ bond, which is supported by the quantum-chemical results. Actinide-actinide bonds is a field of current interest, especially the resulting insights into bonding, reactivity and structure. However, though attracted considerable theoretical interests, solid experimental work on this topic is extremely rare. Thus, these results are of fundamental importance to the actinide chemistry and met the requirement of originality, significance and impact required by Nat. Commun. The paper is well written, and the experimental work is also complemented by computational studies. Here I recommend its publication in the journal after the following concerns are additionally addressed.

1. Since the Th³⁺-Th³⁺ bond is a sigma bond, Th₂@C₈₀ compound is likely to be diamagnetic. 13 NMR spectrum would be informative in this case to determine the diamagnetic nature of the compound.

2. In ref. 14, the authors proposed that U-U bond inside the cage is weak bond. How's difference between the bond energy of U-U bond and Th-Th bond inside Ih fullerene cage? It would be informative for to give a brief comparative discussion between these two kinds of actinide-actinide bond.

3. The authors mentioned in the XAS analysis that single-crystal structure suggests that it likely possesses 12 Ferrocene like Th-C bonds. Can these bonding interactions be observed in the radial distance $\chi(R)$ space spectra of XAS? More details should be given in the ms or SI.

4. The authors should clearly state what functional are used for the structural optimization. And the calculations of the relative energies with BP86, PBE and PBE0 is based on the optimized structure at

PBE0 functional?

5. The authors described that "since no more than 6e are transferred between the guest and the host, the oxidation state of the Th atoms in Th₂@Ih(7)-C₈₀ should be +3", I am very curious why the oxidation state of the Th atom in this complex is not +2? Although they have provided two references to support the point, could they give further comments or try to test the Th(+2)?

6. According to the relative energy of DFT, the triplet $\sigma^1\pi^1$ is the lowest among all the spin state as author state that the energy difference reaches 4 kcal.mol⁻¹. The CASPT2 calculations is used the the singlet optimized structure, it is suggested the author should try to perform CASPT2 calculations based on the triplet $\sigma^1\pi^1$ optimized structure to confirm whether the results are the same.

7. I am very curious that the distance of the Th-Th bond is 6.02Å in D₂(821)-C₁₀₄ and the corresponding EBO still reach 0.76, which may be a very important result. Could authors give further explanation or comment?

8. "DFT" and "EMF" should give full spelling in its first appearance, respectively.

Response to reviewer's comments:

Reviewers' comments:

Reviewer #1 (Remarks to the Author):

The report by Poblet and Chen and their respective co-workers detail the synthesis and characterization of C80-fullerene encaged Th(III)-Th(III) dimer. The experimental works appears to be carried out with care and the characterization of this species by X-ray crystallography, Raman spectroscopy, mass spectrometry and X-ray absorption analyses are consistent with their claims. Computational studies at the DFT and CASPT2 level of theory were employed to gain additional insights into the bonding of this interesting species. Overall it constitutes an interesting contribution to the field of actinide chemistry, which consequently deserves to be published. However, a very similar study on U2@Ih(7)-C80 was reported recently by the same leading authors in JACS in 2017 (ref. 14). Now they expanded this study to Th, and similar investigations concerning the 4f-elements (La, Ce, Dy, etc.) have also been detailed. The advancement herein compared to the previous ones is that Th apparently shows the strongest covalent interaction consistent with a Th-Th bond. Granted this is a novel and interesting finding, and I also admit that low-valent Th(III) species are rare, but they are not without precedent. The same holds true for Th(II) and U(II) species as correctly referenced by the authors. As a synthetic inorganic chemistry, I am also a little bothered about the significance of a reaction that yields 1 mg of a pure compound. Synthetically this is not a useful quantity and hampers any reactivity study which would be of significant interest to the community.

In conclusion, while I would support publication of this contribution with respect to technical quality, I am not convinced that this paper carries enough importance and novelty to justify its publication in Nature Communications.

Some additional scientific requests:

1.) Can the singlet ground state be confirmed by magnetic susceptibility studies?

Response: Thanks for the reviewer's suggestion. It is always highly challenging to conduct magnetic susceptibility studies on these actinide fullerene samples. And due to

the Covid-19 pandemic, we were unable to conduct the magnetic susceptibility studies with our German collaborators who are specialized with this kind measurement and helped us with previous studies. Nevertheless, we managed to carry a ^{13}C NMR test (see Fig. S9 in the revised Supplementary Information) and it does not show any paramagnetic broaden of the ^{13}C , which could also be an alternative evidence of the singlet ground state of the molecule. The corresponding discussion has been added in the revised manuscript (Page 11, last paragraph).

2.) How thermally stable is this species?

Response: This species is stable in the ambient condition either in solid form or in the solvate status. Current sample was stored in solid form over a year, we haven't observed any chemical change so far.

Reviewer #2 (Remarks to the Author):

The authors of the manuscript present the syntheses and characterisation of two Th atoms enclosed in a fullerene ($\text{Th}_2@Ih(7)\text{-C}_{80}$). They also investigate the nature of the Th-Th bond by applying several modern theoretical approaches. Those analyses appear reasonable but surely a sound theoretician, expert in computations of actinides, should review them. As the authors mention, indeed there are a very few examples in the literature where An-An bond was stabilized and its properties were studied not only theoretically but also experimentally. However, considering that the first paper of the authors about $\text{U}_2@C_{80}$ on basically the same topic was in JACS, I do wonder why this paper on Th should be in Nat. Commun.? The authors do not really present for example any novel spectroscopy method, which maybe provides a deeper inside on the electronic structure of the Th-Th bond. Until the XAS part I was actually still thinking that even if the paper should be in my opinion in a journal with a smaller impact factor due to the lack of novelty, it is still a very nice paper. The actinide community will find it from high interest. However, the X-ray absorption data and analyses are simply wrong. I strongly advise the authors to completely reconsider them. The paper can be only

rejected from any journal in the current form. More specific comments follow:

1) The authors report that the Th XAS spectra in Fig. 4a) are $\chi(E)$. this however look to me normalised $\mu(E)$ spectra. Usually, only the oscillating part of XAS spectra after the main absorption peak are called $\chi(E)/\chi(k)$.

Response: We thank the reviewer for pointing out this error. The previous XAS spectra in Fig. 4a indeed were the normalized $\mu(E)$ ($\mu(E)$) rather than $\chi(E)$ ($\chi(E)$) spectra. We apologized for this mistake. In the revised manuscript, we have remeasured the XAS spectra and now present it as the normalized X-ray absorption near edge structure (XANES) $\mu(E)$ spectra. More details please see Fig. 4a in the revised manuscript in page 14.

2) The XAS spectrum of the $\text{Th}_2@I_h(7)\text{-C}_{80}$ in Fig. 4 has several strange peaks/features at about 16200 eV (one feature) and at about 16600 eV (two features). Those look like artefacts, which are a problem for the analyses of the data.

Response: We thank the reviewer for this comment. To address this problem, we have remeasured the XAS data of $\text{Th}_2@I_h(7)\text{-C}_{80}$. We improved the sample preparation method and tripled the amount of the $\text{Th}_2@I_h(7)\text{-C}_{80}$ sample used for the XAS measurements. The current sample were prepared by filling into the hole with 1.5 mm diameter on PTFE films while previously we placed the sample into the hole with 0.5 mm diameter on iron films. The present data was produced as an average of three consecutive scans for the fullerene sample. (see Supplementary Information for details). The new data, which now has significantly improved signal-to-noise ratio, are shown in Fig. 4 in page 14. The features at about 16200 eV and 16600 eV disappeared in the new XAS spectrum. It suggests that, as the reviewer suspected, these features should be noise or false signals due to the poor quality of the previous data, rather than the characteristic peaks of $\text{Th}_2@I_h(7)\text{-C}_{80}$. We apologized for this mistake and gratefully thank the reviewer for helping us avoid it.

3) The authors suggest that there is “pre-edge” at 16200 without giving any interpretation. This however does not look at all like a spectral feature. The difference pointed out at 16600 is very likely also an artefact (see point 2).

Response: We thank the reviewer for this comment. As explained in response to point 2, we have recollected the XAS data of $\text{Th}_2@I_h(7)\text{-C}_{80}$ and confirms that the features at 16200 eV and 16600 eV in the previous spectrum, which disappeared in the current spectrum, were in fact artefacts. More details please find in point 2 and Fig. 4a in the revised manuscript page 14.

4) Why were the XAS spectra modelled with Gaussian + arc-tangent function? The authors do not explicitly say this.

Response: Thanks reviewer for the question. In the previous version, we modelled XAS spectra with Gaussian and arc-tangent function to determine the edge energy since the inflection point of these arc-tan functions is defined as E_0 according to the reference “*Radioanal. Nucl. Chem.* **2003**, 255, 155-158” in which the authors reported the same fitting method to compare the E_0 of $\text{U}@C_{82}$ and UO_2 . We have realized that the Gaussian and arc-tangent function should be an unconventional method. Here, to be simplified, we have revised the related content in our revised manuscript as following: “the maximum peak position in the first derivative of normalized $\mu(E)$ spectra is defined as E_0 , namely the inflection point energies. In particular, the E_0 found for $\text{Th}_2@C_{80}$, 16306.7 eV, is shifted to lower energy by 4.7 eV compared with that of ThO_2 , 16311.4 eV (see Supplementary Information Fig. S8)” More details are shown in the revised manuscript (Page 12, Paragraph 1).

5) Both FT-EXAFS spectra in Fig. 4 a b look unusual and not correct. The FT-EXAFS of ThO_2 appears to have high noise level. The FT-EXAFS spectrum of $\text{Th}_2@C_{80}$ has one single peak, which looks almost like a Gaussian curve, which has an extremely smooth right slope. The FT-EXAFS spectra do not look like this even if the materials are highly disordered.

Response: We thank the reviewer for this comment and admit that the poor quality of the previous XAS data and lack of experience for such a complicated system led to such inaccuracy and problems in data processing. Thus, we have remeasured the XAS data of $\text{Th}_2@I_h(7)\text{-C}_{80}$ and ThO_2 which have higher signal-to-noise ratio than before. In the new XAS spectrum of $\text{Th}_2@I_h(7)\text{-C}_{80}$, the peaks located around 2.40, 2.92 and 3.37 Å can be attributed to the scattering path of Th-C bond and the signal peak located around 3.92 Å can be assigned as the scattering path of Th-Th bond in $\text{Th}_2@I_h(7)\text{-C}_{80}$, which agrees with their corresponding structural diameter obtained from the single crystal structure. We have revised the related discussion in the revised manuscript in page 12, paragraph 2.

6) The authors claim “subtle peaks” between 2.3 -3 Å corresponding to Th-C bonds. There are no peaks visible at this region, which can be attributed to Th-C bonds.

Response: Thanks for the reviewer’s correction. We recognize that we did have problems with data processing previously and therefore the $\chi(R)$ spectrum of $\text{Th}_2@I_h(7)\text{-C}_{80}$ was incorrect. Thus, we have corrected our analysis of the peaks corresponding to Th-C bonds based on the new XAS data. In the new spectrum, three obvious peaks located around 2.40, 2.92 and 3.37 Å can be attributed to the scattering path of Th-C bond from first to third coordination shell in $\text{Th}_2@I_h(7)\text{-C}_{80}$, which is consistent with ferrocene like Th-C bond observed in single crystal structure. More detailed discussion have been presented in our revised manuscript in page 13.

7) I do not understand what is meant under “different distance states”.

Response: We are very sorry for this confusion. In the previous manuscript, “different distance states” referred to “different coordination shell concept”. We have deleted this item and replace it with “coordination shell concept” in our revised manuscript. Please see the revised manuscript (Page 12, Line 18-20 and 23-24).

8) I have no opinion about the wavelengths analyses due to my limited experience with this type of analyses. But, the fitting of the EXAFS data is simply wrong. The authors

suppose that they see Th-C scattering in the spectrum but do not take it into account in their modelling. They claim that their model is simplified without the Th-C bonds but do model the entire data, which should include this Th-C bonds.

Response: Thanks for the reviewer's suggestion. We are very sorry for the previously wrong fitting of the EXAFS data and the mismatch between the measured spectrum and the modelling pattern. In the remeasured XAS spectrum of $\text{Th}_2@I_h(7)\text{-C}_{80}$, we have taken the scattering path of Th-C bonds into account in the fitting process. The $\chi(\text{R})$ space spectra of $\text{Th}_2@I_h(7)\text{-C}_{80}$ displays Th-C bonding with coordination number (CN) approaching 6.0 at 2.44 Å; 6.0 at 2.968 Å and 8.0 at 3.40 Å respectively from first to third coordination shell. More related results have been presented in our revised manuscript in page 13.

9) It is not given the k range of the data, which was modelled. The structural parameters obtained from the EXAFS fit like coordination numbers, DW factors, S0 and E0 parameters are not given. The data in k range (not filtered) should be given together with the fit.

Response: We thank the reviewer for this suggestion. The modelled k range is 1–11.06 Å⁻¹ for $\text{Th}_2@I_h(7)\text{-C}_{80}$. More details about structural parameters obtained from the EXAFS fit, including the R factor, coordination numbers, S0 and E0 parameters have been provided in Table S8 in our revised Supplementary Information.

Reviewer #3 (Remarks to the Author):

This paper reported an important result: the synthesis and characterizations of a strong actinide-actinide bond, which is long sought after, but hardly accessible by conventional synthetic method so far. This compound was synthesized using the arc-discharge method and fully characterized using several techniques. The structure of $\text{Th}_2@C_{80}$ was unambiguously determined by single-crystal X-Ray diffraction analysis. A in-depth XAS analysis also confirms the presence of the strong covalent Th³⁺-Th³⁺ bond, which is supported by the quantum-chemical results. Actinide-actinide bonds is

a field of current interest, especially the resulting insights into bonding, reactivity and structure. However, though attracted considerable theoretical interests, solid experimental work on this topic is extremely rare. Thus, these results are of fundamental importance to the actinide chemistry and met the requirement of originality, significance and impact required by Nat. Commun. The paper is well written, and the experimental work is also complemented by computational studies. Here I recommend its publication in the journal after the following concerns are additionally addressed.

1. Since the Th³⁺-Th³⁺ bond is a sigma bond, Th₂@C₈₀ compound is likely to be diamagnetic. ¹³C NMR spectrum would be informative in this case to determine the diamagnetic nature of the compound.

Response: Thanks for the reviewer's suggestion. We've carried out the ¹³C NMR and the results are shown in Fig. S9 in the revised Supplementary Information and the corresponding text has been added at the bottom of page 11.

2. In ref. 14, the authors proposed that U-U bond inside the cage is weak bond. How's difference between the bond energy of U-U bond and Th-Th bond inside Ih fullerene cage? It would be informative for to give a brief comparative discussion between these two kinds of actinide-actinide bond.

Response: Thanks for the reviewer's suggestion. Although we are preparing an article describing under what conditions strong covalent interactions between U atoms can be obtained, we have added a brief comment in the revised manuscript (Page 10), related to point 7.

3. The authors mentioned in the XAS analysis that single-crystal structure suggests that it likely possesses 12 Ferrocene like Th-C bonds. Can these bonding interactions be observed in the radial distance $\chi(R)$ space spectra of XAS? More details should be given in the ms or SI.

Response: Thanks for the reviewer's suggestion. Since the poor quality of previous

XAS spectra results in problems in data processing and analyses, we have remeasured and obtained a new XAS data with better quality. In the current XAS spectrum of $\text{Th}_2@I_h(7)\text{-C}_{80}$ with higher signal-to-noise ratio, three obvious peaks located around 2.40, 2.92 and 3.37 Å can be attributed to the scattering path of Th-C bond (Fig. 4b and 4d in page 14), which is consistent with ferrocene like Th-C bond observed in single crystal structure. More details are shown in the revised manuscript in page 12.

4. The authors should clearly state what functional are used for the structural optimization. And the calculations of the relative energies with BP86, PBE and PBE0 is based on the optimized structure at PBE0 functional?

Response: Thanks for the reviewer's suggestion. All DFT energies correspond to fully optimized geometries in each functional. For CASPT2 calculations, we have used PBE0 geometries. A comment on this point is added in the Computational Methods section.

5. The authors described that “since no more than 6e are transferred between the guest and the host, the oxidation state of the Th atoms in $\text{Th}_2@I_h(7)\text{-C}_{80}$ should be +3”, I am very curious why the oxidation state of the Th atom in this complex is not +2? Although they have provided two references to support the point, could they give further comments or try to test the Th(+2)?

Response: Cage $I_h(7)\text{-C}_{80}$ is observed many times in endofullerene chemistry because it is able to accept 6e from the internal guest (Th_2 in the present case) and create a large HOMO-LUMO gap that confers stability to the endofullerene. On the other hand, Th tends to behave as +4, therefore, it is hard to believe that the actinide could behave a +2 ion. It is also important to emphasize that the calculations do not a priori presume any particular oxidation state and it is the resulting electron distribution that confirms the nature of the ground state. In the present case the unique occupied MO orbital with a dominant population of valence shell Th orbitals is the sigma-type Th-Th bonding orbital. All the other valence shell Th orbitals are unoccupied and this is the element that defines the nature of the formal oxidation state of the metal. A brief comment was

included in the revised manuscript.

6. According to the relative energy of DFT, the triplet $\sigma 1\pi 1$ is the lowest among all the spin state as author state that the energy difference reaches 4 kcal.mol⁻¹. The CASPT2 calculations is used the the singlet optimized structure, it is suggested the author should try to perform CASPT2 calculations based on the triplet $\sigma 1\pi 1$ optimized structure to confirm whether the results are the same.

Response: This energy has already been included in the previous version of the manuscript, entry 6-footnote c in Table 1, and briefly mentioned in page 6-7.

7. I am very curious that the distance of the Th-Th bond is 6.02Å in D2(821)-C104 and the corresponding EBO still reach 0.76, which may be a very important result. Could authors give further explanation or comment?

Response: This is the result of a very high overlap between 7s/6d orbitals of actinides, as shown in Fig S6. This property is used by Th atoms to stabilize two electrons in a sigma-type bonding orbital, even at very long distances. Does it mean that it will be easy to synthesize very large Th endofullerenes? We believe that in the future, cages with more than 90 carbon containing two Th atoms will likely be isolated. Meanwhile, making use of this property, we have isolated and characterized mixed cages containing an actinide and a lanthanide, for which we have characterized a metal-metal bond with only one-electron at distances > 4 Å (manuscript in preparation). We are persuaded that this property could also be used in more traditional chemistry, as for example in organometallic chemistry, to design ligands to favor interaction at large distance between actinide (+3) ions. We have added a brief comment in page 10.

8. “DFT” and “EMF” should give full spelling in its first appearance, respectively.

Response: We thank the reviewer for the kind suggestion. Now we have given the full spelling of both terms in the introduction of the revised manuscript.

REVIEWER COMMENTS

Reviewer #3 (Remarks to the Author):

In my opinion, the authors have carefully responded the concerns raised by reviewers. It's my pleasure to support publication of this manuscript in its current form.

Reviewer #4 (Remarks to the Author):

In accordance with a request from the editorial office, I have reviewed the revised manuscript especially for the XAS measurement and analysis for Th₂@Ih(7)-C80. I indeed feel this work is interesting, and would be eligible to be published from Nat. Commun. However, the XAS part is still needed to be considered much more and corrected appropriately. I'm wondering the authors are not very familiar with XAS. As long as I read the supplementary information (SI), descriptions about XAS experiments (page 4) are rather incomplete, because there are many errors and mistakes. At this status, it is quite difficult to make readers understand correctly. I do not know why k²-weight in the EXAFS analysis is employed here. The k³-weight will provides much better resolution, and therefore it is much more popular. Data presentations are also confusing. In Figure 4b, the x-axis is actually "R (Å)"? Usually, the R-space tends to be displayed without any correction of phase shift. Therefore, the R-space should be drawn as a function of "R + Δ (Å)", and called "radial structure function". Related to this point, the best fit R_s reported in Table S8 do not match with those described in the main text. This is serious problem, because I cannot know what is correct. I do not understand why the x-axis of Fig. 4b does not start from 0, but does from 1. I also cannot find any importance of ThO₂ in this study. I'm aware that ThO₂ has been employed as a reference material. But if so, it should be simply kept in SI. I strongly recommend to describe more details about relationship between peaks assigned to the first-third Th-C interactions observed in Fig. 4b and actual molecular structure of this compound determined by SCXRD. The first Th-C interaction can be connected to Fig. 1, but this is not the case for the second and third Th-C interactions. The EXAFS fit results in the main text are also quite confusing, because CNs described here seem to be described with significant digits like 6.0, which means estimated CN is ranging from 5.5 to 6.4. In contrast, the analysis procedure shown in SI says "CNs were fixed as the nominal values". Which is correct? First of all, what is "nominal value"? In Fig. 4d, the third Th-C interaction is not well reproduced in the fit. Why?

Response to reviewer's comments:

Reviewers' comments:

Reviewer #4:

In accordance with a request from the editorial office, I have reviewed the revised manuscript especially for the XAS measurement and analysis for Th₂@I_h(7)-C₈₀. I indeed feel this work is interesting, and would be eligible to be published from Nat. Commun. However, the XAS part is still needed to be considered much more and corrected appropriately.

1. I'm wondering the authors are not very familiar with XAS. As long as I read the supplementary information (SI), descriptions about XAS experiments (page 4) are rather incomplete, because there are many errors and mistakes. At this status, it is quite difficult to make readers understand correctly. I do not know why k²-weight in the EXAFS analysis is employed here. The k³-weight will provide much better resolution, and therefore it is much more popular.

Reply: Many thanks for the reviewer's suggestive comments. We have to admit that it is indeed the first time we perform the EXAFS analyses for such a complicated system, which is very challenging for us. We agree with the reviewer that k³-weight is commonly used in the EXAFS analysis and usually provides better resolution. The reason we use k²-weight $\chi(R)$ in this particular case is that it would make the scattering path of Th-C bond interaction from second to third coordination shell more recognizable. Following reviewer's suggestion and for comparison, we carried out the k³-weight $\chi(R)$ and now present it in the revised Supplementary Fig. S10 (Page 14).

2. Data presentations are also confusing. In Figure 4b, the x-axis is actually "R (Å)"? Usually, the R-space tends to be displayed without any correction of phase shift. Therefore, the R-space should be drawn as a function of "R + Δ (Å)", and called "radial structure function". Related to this point, the best fit R_s reported in Table S8 do not match with those described in the main text. This is serious problem, because I cannot know what is correct.

Reply: We thank the reviewer for this guidance and correction. Following this suggestion, we revised the R-space and the corresponding fitting curves which are now presented as the radial structure function of “R + ΔR ” (Fig. 4b, 4d) and ΔR (+0.41) is determined by the difference between the first shell Th-C distance from single crystal structure (2.48) and the first scattering path (2.07). Accordingly, we have revised the EXAFS fitting part description in the Supplementary Information as following: “All fits were performed in the radial structure function of “R + ΔR ” with k-weight of 2.” We have also revised the fit R values in Supplementary Table S8 (ESI, Page 19) according to the revised data of R-space and they are now consistent with the corresponding discussion in the main text.

3. I do not understand why the x-axis of Fig. 4b does not start from 0, but does from 1.

Reply: Previously we presented the R-space curve (Fig. 4b) with the x-axis start from 1 because we considered that there should be no bond under 1 Å. But we have recognized that it was not rigorous. We thank the reviewer for this correction. Following this comment, we now present the x-axis starts from 0 while ΔR is added. Thus, the curve will start from $0+\Delta R$ leaving the blank from 0 with ΔR value. More details please see revised Fig. 4b in the revised manuscript page 14.

4. I also cannot find any importance of ThO₂ in this study. I’m aware that ThO₂ has been employed as a reference material. But if so, it should be simply kept in SI.

Reply: We thank the reviewer for this comment and as suggested, we moved the spectrum of ThO₂ into the Supplementary Information as Fig. S11 (Page 14) and deleted the corresponding discussions.

5. I strongly recommend to describe more details about relationship between peaks assigned to the first-third Th-C interactions observed in Fig. 4b and actual molecular structure of this compound determined by SCXRD. The first Th-C interaction can be connected to Fig. 1, but this is not the case for the second and third Th-C interactions.

Reply: We thank the reviewer for this suggestion. Following this suggestion, we add

the discussion of the connection between second and third Th-C interactions and the crystallographic structure as following: “Besides the relative strong first Th-C interactions, the observation of the second and third Th-C interaction suggests that Th ions may also have weaker but non-negligible interaction with carbon atoms in wider region. As shown in Fig. S12 and S13, the Th-C distances of 2.90 Å and 3.35 Å (average values) can be found in the crystal structure of $\text{Th}_2@I_h(7)\text{-C}_{80}$, which correspond to the second and third Th-C interaction peaks at 3.11 and 3.54 Å, respectively.” (Please see the revised manuscript page 12-13) Accordingly, we add the corresponding figure as Fig. S12 and S13 at Page 15-16, ESI.

6. The EXAFS fit results in the main text are also quite confusing, because CNs described here seem to be described with significant digits like 6.0, which means estimated CN is ranging from 5.5 to 6.4. In contrast, the analysis procedure shown in SI says “CNs were fixed as the nominal values”. Which is correct? First of all, what is “nominal value”?

Reply: Thank the reviewer for this comment and we are sorry for the confusion and the use of unprofessional terms. The term “nominal value” intended to describe the coordination numbers according to its single crystal structure but is not professional. We have deleted it in the revised Supplementary Information (Page 5). Actually, the coordination numbers of Th in the fullerene cage can only be considered as within a certain range and could not be determined as a ‘fixed’ number, since the current theoretical studies show that interaction between Th and the fullerene cage is rather complicated. The text “CNs were fixed as the nominal values” in previous Supplementary Information was wrong. We apologized for this mistake and deleted it in the revised version.

7. In Fig. 4d, the third Th-C interaction is not well reproduced in the fit. Why?

Reply: We thank the reviewer for this comment and rechecked our fitting method. We found that the poor reproducibility of the third Th-C interaction may be the results of the fixed second and third coordination shell fitting parameters. Thus, we revised the

fitting parameter and keep each S_0^2 for the Th-C interaction from first to third coordination shell running freely. It turned out that the current results show a lower R factor and the third Th-C interaction fitting become more reasonable. More details please see the revised Fig. 4d (revised manuscript page 14) and supplementary text (revised ESI page 5).

REVIEWERS' COMMENTS

Reviewer #4 (Remarks to the Author):

Thank you for your response. I'm happy that most of my concerns have been sufficiently addressed in this revision. But, please allow me to make further minor comments.

Regarding my 2nd point, the R-space was shifted by +0.41 Å (Å: Angstrom sign) in this revision. This means that the phase shift (ΔR) has been included. So, now you should call Figs. 4b and 4d as "radial distribution functions", and their the x-axes should be named as "R /Å". Related to this, the description at 9th line from the bottom of p.12 of the main text should be corrected as follows.

"radial distribution function including the phase shift (+0.41 Å) is..."

In my 5th concern, it is still difficult to claim similarity between Th-C distances found in SCXRD (2.90 and 3.35 Å) and EXAFS (3.11 and 3.54 Å) at a glance. I'm wondering that such a difference would be ascribed to the difference in the experiment temperatures, i.e., SCXRD at 113 K, while EXAFS at RT, right? If so, you should make such a comment around the bottom of p.12.

If all above points are fully addressed, I recommend this manuscript for publication from Nat. Commun.

Response to reviewer's comments:

Reviewers' comments:

Reviewer #4:

Thank you for your response. I'm happy that most of my concerns have been sufficiently addressed in this revision. But, please allow me to make further minor comments.

Regarding my 2nd point, the R-space was shifted by +0.41 Å (Å: Angstrom sign) in this revision. This means that the phase shift (ΔR) has been included. So, now you should call Figs. 4b and 4d as "radial distribution functions", and their the x-axes should be named as "R / Å". Related to this, the description at 9th line from the bottom of p.12 of the main text should be corrected as follows.

"radial distribution function including the phase shift (+0.41 Å) is..."

Reply: We thank the reviewer for this suggestion. Accordingly, we have revised Fig. 4b and 4d as "radial distribution functions" and the x-axes have been named as "R / Å". We have also revised the corresponding description as following: "The radial distance space spectrum $\chi(R)$, obtained by the Fourier transform of the Th L₃-edge EXAFS oscillation, $\chi(k)$ of Th₂@I_h(7)-C₈₀ (k range 2.54-10.42 Å⁻¹), is presented as the radial distribution function including the phase shift (+0.41 Å). This shift is determined by the difference between the first shell Th-C distance from single crystal structure (2.48 Å) and the first scattering path (2.07 Å)." in the main text Page 12.

In my 5th concern, it is still difficult to claim similarity between Th-C distances found in SCXRD (2.90 and 3.35 Å) and EXAFS (3.11 and 3.54 Å) at a glance. I'm wondering that such a difference would be ascribed to the difference in the experiment temperatures, i.e., SCXRD at 113 K, while EXAFS at RT, right? If so, you should make such a comment around the bottom of p.12.

Reply: We thank the reviewer for the suggestive comments. Accordingly, we have added a comment at the end of the first paragraph in Page 13 as following: "Note that the differences between Th-C distances found in SCXRD and EXAFS might be ascribed to different methodologies as well as the different measure temperatures, as

the single crystal X-ray data were collected at 113 K and the XAS experiment was carried out at room temperature (ca. 298 K).”